# Gamma-Glutamyltransferase Is a Predictor for Future Changes of Diabetogenic Factors in Aged Chinese—A Four-Year Follow-Up Study

**DOI:** 10.3390/jcm12175606

**Published:** 2023-08-28

**Authors:** Man Sze Wong, Chun Yen Jun Lo, Yen-Lin Chen, Fang-Yu Chen, Chun-Heng Kuo, Jin-Shuen Chen, Dee Pei, Pietro Pitrone, Chung-Ze Wu

**Affiliations:** 1Department of Medicine, School of Medicine, Fu Jen Catholic University, New Taipei City 24352, Taiwan; mancycat1@gmail.com (M.S.W.); junlocy@gmail.com (C.Y.J.L.); peidee@gmail.com (D.P.); 2Department of Pathology, Tri-Service General Hospital, National Defense Medical Center, Taipei City 11490, Taiwan; anthonypatho@gmail.com; 3Division of Endocrinology and Metabolism, Department of Internal Medicine, Fu Jen Catholic University Hospital, School of Medicine, College of Medicine, Fu Jen Catholic University, New Taipei City 24352, Taiwan; julia0770@yahoo.com.tw (F.-Y.C.); cpp0103@gmail.com (C.-H.K.); 4Kaohsiung Veterans General Hospital, Kaohsiung City 81362, Taiwan; dgschen@vghks.gov.tw; 5Institute of Precision Medicine, National Sun Yat-sen University, Kaohsiung City 80424, Taiwan; 6Division of Nephrology, Department of Medicine, Tri-Service General Hospital, National Defense Medical Center, Taipei City 11490, Taiwan; 7Radiology Department, Papardo Hospital, 98100 Messina, Italy; pieropitrone@live.it; 8Division of Endocrinology and Metabolism, Department of Internal Medicine, School of Medicine, College of Medicine, Taipei Medical University, Taipei City 11031, Taiwan; 9Division of Endocrinology and Metabolism, Department of Internal Medicine, Shuang Ho Hospital, Taipei Medical University, New Taipei City 23561, Taiwan

**Keywords:** type 2 diabetes mellitus, gamma-glutamyl transferase, first phase and second phase insulin secretion, glucose effectiveness

## Abstract

Glucose homeostasis in the body is determined by four diabetes factors (DFs): insulin resistance (IR), glucose effectiveness (GE), and the two phases of insulin secretion—first phase (FPIS) and second phase (SPIS). Previous research points to a correlation between elevated levels of gamma-glutamyl transferase (γGT) and an increased risk of type 2 diabetes. This study investigates the relationship between γGT and the four DFs in older Chinese individuals. This study involved 2644 men and 2598 women, all of whom were relatively healthy Chinese individuals aged 60 years or more. The DFs were calculated using formulas developed by our research, based on demographic data and factors related to metabolic syndrome. Pearson’s correlation was utilized to assess the relationship between γGT and the four DFs. The findings suggested a positive correlation between γGT and IR, FPIS, and SPIS, but a negative correlation with GE in men. Among women, only SPIS and GE were significantly correlated with γGT. The factors showed varying degrees of correlation, listed in descending order as follows: GE, SPIS, FPIS, and IR. This study confirms a significant correlation between γGT and DFs in this population, highlighting the noteworthy role of GE.

## 1. Introduction

The global incidence of type 2 diabetes (T2D) has experienced a significant increase over the last two decades, a pattern that has also clearly been observed in Taiwan. [1] Current estimates suggest that more than two million Taiwanese people are living with T2D, and the annual rate is projected to rise by an extra 25,000 individuals. [1] Currently, T2D ranks as the fifth leading cause of mortality in Taiwan, taking nearly 10,000 lives each year. The significant medical burden and complications associated with T2D are substantial and cannot be underestimated. At the same time, Taiwan features in the top three countries worldwide with the highest rate of aging. The highest incidence of T2D in Taiwan occurs among the senior population, those aged over 65, who represent more than 40% of Taiwan’s total T2D patient population [1].

Although extensive research has been conducted, the complex pathophysiology of T2D is only partially understood. Nevertheless, it is widely accepted that insulin resistance (IR) and impaired insulin secretion are key contributors [2]. Of particular interest are the two stages of insulin secretion: the first phase of insulin secretion (FPIS) and the second phase of insulin secretion (SPIS) [3]. The FPIS represents the acute insulin response in the initial 10 mins following glucose intake, whereas the SPIS extends for 2–3 h after the FPIS. A fourth critical, but often neglected, component is glucose effectiveness (GE)—the capacity of glucose to enhance its own utilization and inhibit hepatic glucose production in post-meal states [4]. These four elements are collectively referred to as ‘diabetes factors’ (DFs) in the scope of our current study.

Gamma-glutamyl transferase (γGT), an enzyme primarily found on the exterior surfaces of most cells—especially in the liver—serves a significant metabolic function as a transport molecule in the metabolism of various drugs and toxins [5]. Typically, increased levels of γGT are indicative of liver disease and have been found in higher quantities in patients with T2D [6]. Building on this observation, Lee et al. recognized an increase in γGT within its physiological range as a sensitive, early biomarker for the onset of T2D [7]. This significant correlation could be attributed to the relationship between IR and fatty liver disease, as impaired glucose metabolism can result in heightened γGT levels [8]. However, to our knowledge, there have been no prior studies investigating the relationships between γGT and the other three DFs. In our current study, we aim to investigate the relationship between γGT and all four DFs within a group of older Chinese individuals, observed over a span of four years.

## 2. Materials and Methods

### 2.1. Study Subjects

Initially, there were 9826 subjects, aged between 60 and 87 years, recruited from the MJ Health Screening Center, Cardinal Tien Hospital, and the Tri-Service General Hospital in Taiwan over a four-year period. Data were collected anonymously during routine health check-ups once informed consent had been obtained from the participants. Any individuals with diagnosed liver diseases, including liver cirrhosis, alcoholic hepatitis, or chronic hepatitis B or C, were excluded from this study. However, those with non-alcoholic fatty liver diseases were only excluded if their AST/ALT levels exceeded three times the upper limit. In addition, those taking medication affecting blood pressure, glucose, or lipid levels were not included in this study. Finally, this study enrolled 5242 participants (2644 men and 2598 women) (Figure 1). Each participant provided written consent to participate in this research. The research project protocol received approval from the relevant ethics committees of both the Tri-Service General Hospital (Approval No: TSGH-100-05-246) and Cardinal Tien Hospital (Approval No: CTH-100-2-5-036).

### 2.2. Anthropometry and Laboratory Assessments

Body Mass Index (BMI, kg/m^2^) was calculated by dividing the participant’s weight in kilograms (kg) by the square of their height in meters (m). Blood pressure, both systolic and diastolic (SBP and DBP, respectively, measured in mmHg), was assessed by a nurse using a standard mercury sphygmomanometer, specifically the Citizen CH-5000 model (Citizen, Tokyo, Japan), on the right arm of the seated participant. The waist circumference (in cm) was measured with a tape measure, positioned midway between the lower rib cage margin and the top of the hip bone (ilium) around the subject’s abdomen.

Blood samples were obtained from the antecubital vein following a fasting period of 10 h for a comprehensive blood count analysis. Fasting plasma glucose (FPG) measurements were carried out using plasma via the glucose oxidase method (utilizing a YSI 203 glucose analyzer, Yellow Springs Instruments, Yellow Springs, USA), and lipid profiles (inclusive of triglycerides, TG) were assessed using the Fuji DriChem 3000 analyzer (Fuji Photo Film, Tokyo, Japan). Through enzymatic cholesterol assay post dextran sulfate precipitation, serum was analyzed to determine the concentrations of high-density lipoprotein cholesterol (HDL-C, mmol/L) and low-density lipoprotein cholesterol (LDL-C, mmol/L). Measurements of γGT were executed as per the Szasz Method, employing a Beckman Coulter AU-5821 instrument (from Shizuoka-ken, Mishima, Japan).

In order to calculate the DFs, we utilized formulas developed by our team, as detailed below (in international units). Herein, we provide a brief demonstration of the validity of these formulas. In these studies, approximately 70% of the participants contributed to the development of the formulas, with the remaining 30% serving as an external validation group. Consequently, this methodology allowed for the verification of the accuracy of the formulas.

(1)IR: There were 327 subjects without T2D enrolled. IR was measured by an insulin suppression test. The r value between the measured and calculated values was 0.581 (*p* < 0.001). The IR data were derived using a formula applied to subjects with normal glucose tolerance in Model 1.

IR = log (1.439 + 0.018 × sex − 0.003 × age + 0.029 × BMI − 0.001 × SBP + 0.006 × DBP + 0.049 × TG − 0.046 × HDL-C − 0.0116 × FPG) × 10^3.333^ [9].

(2)FPIS: There were 186 subjects, self-referred for diabetes screening, enrolled. The FPIS was measured by frequently sampled intravenous glucose tolerance tests. The r value between the measured and calculated values was 0.671 (*p* < 0.001).

FPIS = 10^(1.47−0.119×FPG+0.079×BMI−0.523×HDL-C)^ [10].

(3)SPIS: There were 82 participants, including those with normal glucose tolerance, pre-diabetes, and T2D without oral anti-diabetic agents in the out-patient clinic. The SPIS was measured by a modified low dose glucose infusion test. The r value between the measured and calculated values was 0.65 (*p* = 0.002).

SPIS = 10^(−2.4−0.088×FPG+0.072×BMI)^ [11].

(4)GE: There were 227 participants, including individuals with normal glucose tolerance, pre-diabetes, and T2D who are not taking oral anti-diabetic medications. GE was measured by frequently sampled intravenous glucose tolerance tests. The r value between the measured and calculated values was 0.43 (*p* = 0.001).

GE = (29.196 − 0.103 × age − 2.722 × TG − 0.592 × FPG) × 10^−3^ [12].

### 2.3. Statistical Analysis

The *t*-test was utilized to assess the differences between the two groups. A *p*-value less than 0.05 from a two-sided t-test was deemed to indicate statistical significance. The connections between γGT and the DFs were examined using Pearson’s correlation. All statistical analyses were executed using the SPSS statistical software (version 19.0, IBM Inc., Armonk, NY, USA). The data are displayed as the mean ± standard deviation (SD). To contrast the strength of the four correlation lines, they are illustrated within the same graph, with a steeper slope indicating of a stronger relationship. The comparison of each correlation line was made using Chris’s calculator. [13] Given that only GE has a negative correlation with γGT (in quadrant 4), we drew a reciprocal line to facilitate comparison with the other lines (in quadrant 1).

## 3. Results

Table 1 presents the clinical characteristics of the 5242 participants in this study, comprising 2644 men and 2598 women. This table includes their baseline demographic information and biochemical data. The data reveals that men displayed elevated values in age, DBP, waist circumference, γGT, creatinine, and uric acid compared to women. Conversely, women had a higher BMI, SBP, HDL-C, and LDL-C.

Table 2 illustrates the calculated DFs for both men and women at the end of the follow-up period. Notably, women presented with higher values for SPIS, IR, and GE compared to men. However, the variations in the FPIS were not significant between the two groups.

Table 3 illustrates the direct correlations between γGT and the four DFs. For men, there was a significant positive correlation between γGT and FPIS, SPIS, and IR, while a negative correlation was observed with GE. Conversely, the outcomes for women varied; there were no significant correlations between γGT and FPIS or IR. SPIS exhibited a positive correlation and GE displayed a negative correlation with γGT in women. In the case of men, GE held the strongest correlation with γGT, succeeded by SPIS, FPIS, and lastly, IR. Among women, GE also registered the steepest slope value, followed by SPIS.

Figure 2 illustrates scatter plots correlating γGT with each DF, offering a detailed view of the individual relationships. Figure 3 visually encapsulates the main findings of this study, serving as a graphical display of the data presented in Table 3 (Panel A: for men, Panel B: for women). The difference in significance between the two correlations is visualized by the gap between the two lines. For men, the slope of the reciprocal GE line is notably higher than those of SPIS and IR. Notably, the slope lines for SPIS and IR coincide, but both stand significantly higher than that for FPIS, which registers the least steep slope. Similarly, for women, the slope of the reciprocal GE line is higher than that for SPIS. Since no significant correlations were found between γGT and either IR or FPIS for women, these lines were not included in the figure.

## 4. Discussion

The association between γGT and the incidence of T2D is widely recognized. This investigation stands out as the only longitudinal study that further clarifies the pathophysiological links between these elements in a group of healthy older Chinese individuals. Our results point to a positive correlation between insulin secretion and IR with γGT, while a negative correlation was noted with GE across both genders. In the case of women, however, the associations between γGT, FPIS, and IR did not reach statistical significance. Furthermore, we ranked the strength of these associations in descending order as GE, SPIS, FPIS, and IR. Despite this, the relationships between γGT and both FPIS and IR in women lacked statistical significance.

IR describes the diminished sensitivity to insulin in cells that typically respond to insulin, including those in the liver, muscle, and adipose tissues. In our study, a positive association was discovered between γGT and IR, specifically in men. This finding is not unique and aligns with similar research conducted in middle-aged French populations [14]. The most compelling evidence stems from a longitudinal Korean study led by Ryoo, et al. [15]. Their research employed the homeostasis model assessment to measure IR, following a 3.54 person-years follow-up. It was found that when adjustments were made for age, BMI, alanine aminotransferase, LDL-C, IR, serum creatinine, smoking, alcohol consumption, exercise habits, and hypertension, γGT emerged as a significant predictor of future IR. However, the exact mechanism linking γGT and IR remains elusive. There are two potential explanations. Firstly, γGT has been linked to fatty liver, which is also associated with inflammation and IR [16], suggesting that fatty liver may serve as a bridge between γGT and IR. Secondly, oxidative stress could be another intermediary. Evidence from the Framingham Offspring study indicates that oxidative stress positively correlates with IR [17]. Furthermore, γGT also exhibits a relationship with oxidative stress [16]. Interestingly, this adverse effect could be mitigated by antioxidants [18]. Hence, our study further corroborates the relationship between γGT and IR in elderly Chinese men.

The homeostasis of glucose metabolism principally relies on two pivotal mechanisms: insulin activity and insulin secretion. Data from this current research suggests a notable positive correlation between γGT and both FPIS and SPIS in men and women, with the exception of FPIS in women. As far as we are aware, the most relevant study was conducted by Succurro, et al. [19]. Using an oral glucose tolerance test, they found that there was a correlation between 1 h plasma insulin levels after glucose loading and γGT (r = 0.15, *p* < 0.0001). It is worth noting that these insulin levels can be seen as analogous to the SPIS, since the peak of the FPIS should only occur around 10–15 min [20]. The rationale behind these results is not difficult to understand. As previously indicated, γGT serves as a marker for fat accumulation in the liver, which is closely tied to improved insulin secretion [21,22]. This study contributes to our understanding of the influence of γGT on future insulin secretion alterations, an area that has been largely unexplored until now.

Compared with insulin secretion, there have been a limited number of studies investigating the relationship between γGT and GE. Clinically overt diabetes only occurs when both insulin action and secretion deteriorate to such a degree that glucose homeostasis cannot be maintained. At this point, the sole remaining factor to control glucose levels is GE. Consequently, subjects with more efficient GE may maintain lower plasma glucose levels. The relationship between γGT and GE might be influenced by free fatty acid (FFA), as Zhang and colleagues have demonstrated that FFA is indicative of higher fat storage in the liver [23], which is also associated with higher γGT levels [21,22]. Although a direct cause-and-effect relationship can not be definitively established, a link between γGT and FFA is probable. Reinforcing this idea, a study by Hawkins, et al. establishes an association by demonstrating that FFA has the potential to negatively affect both overall and liver-specific GE [24], thereby creating an essential link between γGT and GE. Further substantiating this, research by Kishore, et al. identified that this adverse impact on GE could be reversed through the reduction of FFA levels [25]. Even though the supporting evidence is limited, we assert that the relationship between γGT and GE demands further investigation.

In recent years, studies have shown that the allostatic load model is linked to the development of T2D, which may explain our findings [26,27]. Allostatic load refers to the cumulative physiological wear and tear experienced by the body due to repeated or chronic stress. Prolonged stress leads to elevated cortisol, a hormone that, when persistently high, disrupts metabolic functions, resulting in IR. Elevated cortisol levels increase glucose production, and over time, with sustained stress and consistently high cortisol, the body’s sensitivity to insulin diminishes. In addition, with chronic stress and the consistent release of these stress hormones, GE could be compromised. The continuous production of glucose by the liver, combined with reduced cellular uptake, can lead to elevated blood glucose levels. Notably, a study by Macit MS, et al. observed that administering hydrocortisone to rats led to a 2.6-fold increase in γGT activity [28]. Moreover, stress-induced behaviors, such as inactivity, further contribute to IR. Additionally, inflammation, heightened by chronic stress, is also linked to IR. Thus, the cycle of stress, allostatic load, and IR highlights the significant role of stress management in preventing conditions like T2D.

A statistically significant association was observed between γGT and the four DFs in elderly men. Nevertheless, the statistical analysis did not reveal any significant relationships between γGT and both FPIS and IR in aged women. The differences between older men and women can be attributed to hormonal changes, fat distribution, and other factors. As they age, men typically experience a decline in testosterone levels, while women see a drop in estrogen levels. However, these levels do not usually reach the low levels observed in the opposite sex. Both estrogen and testosterone are hormones known to influence IR. Moreover, fat distribution plays a role in IR. Women tend to store fat subcutaneously, while men are more likely to accumulate visceral fat around the liver. Other factors, such as lifestyle and genetic predisposition, also contribute to IR and FPIS in both sexes.

Despite the findings, our study does present certain limitations. Firstly, our conclusions are derived from clinical observations and do not incorporate support from animal or cellular studies that could offer stronger evidence in terms of signaling pathways. Secondly, one might argue that DFs were not directly assessed by more sophisticated methods like the intravenous glucose tolerance test. However, implementing such methods within a large cohort of nearly 5000 subjects would be both cost-prohibitive and labor-intensive. In contrast, the substantial sample size could potentially offset any inaccuracies in the measurements of the DFs. Thirdly, γGT is linked to non-alcoholic fatty liver and steatohepatitis. This correlation suggests that γGT could be more than just a casual marker; it may indeed serve as a predictive biomarker, indicating the early stages of fatty liver development and subsequent IR, which might be associated with our four DFs. Unfortunately, we lacked ultrasonography or fibroscan data to confirm the presence of non-alcoholic fatty liver and steatohepatitis in these subjects. Nonetheless, the primary objective of this study was to longitudinally examine the relationship between γGT and the four DFs. Despite the absence of this evidence, our results remain unaffected. Fourth, we utilized components of metabolic syndrome to assess the relationship between γGT and the four DFs. However, we cannot over-interpret the prediction of T2D in our results. Numerous factors, including of family history and physical exercise, also contribute to the development of T2D. It is essential to acknowledge that certain individuals diagnosed with T2D may have blood pressure and BMI levels within the normal range. Lastly, this study was conducted on older Chinese individuals. Therefore, caution must be exercised when attempting to extrapolate this data to other age groups or ethnicities.

## 5. Conclusions

In conclusion, based on our four prior formulas, we can deduce the four DFs to predict the probable advancement and probability of developing T2D in older individuals. During a four-year follow-up period, γGT demonstrates a positive correlation with IR, FPIS and SPIS, and a negative association with GE in elderly Chinese men. In contrast, for women, the correlations were significant only with SPIS and GE. The intensity of these relationships, ranked from the strongest to the weakest, was as follows: GE, SPIS, FPIS, and IR. The role of GE in this context should not be underestimated. In addition, γGT is also considered an inflammatory biomarker. Our findings suggest that if an individual exhibits elevated levels of γGT, it could indicate an impairment in one of the four DFs, potentially predicting the development of T2D or non-alcoholic fatty liver. Moreover, these results may further our understanding of the mechanisms and pathophysiology of T2D, revealing complexities beyond our current comprehension.

## Figures and Tables

**Figure 1 jcm-12-05606-f001:**
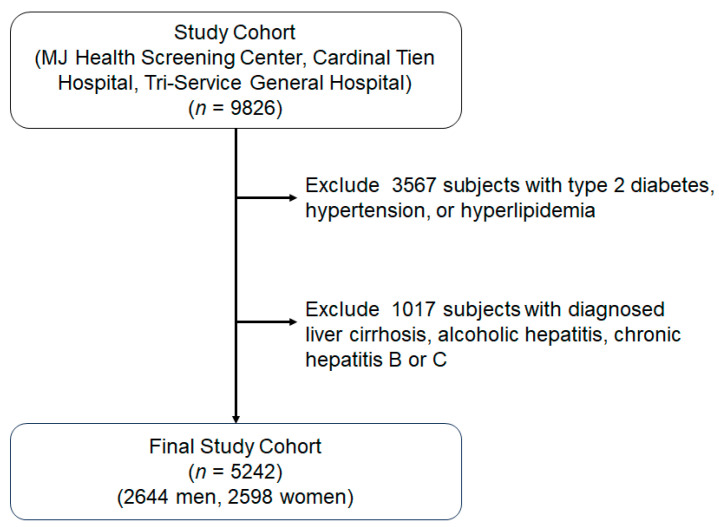
Flowchart of sample selection from MJ Health Screening Center, Carinal Tien Hospital and Tri-Service General Hospital.

**Figure 2 jcm-12-05606-f002:**
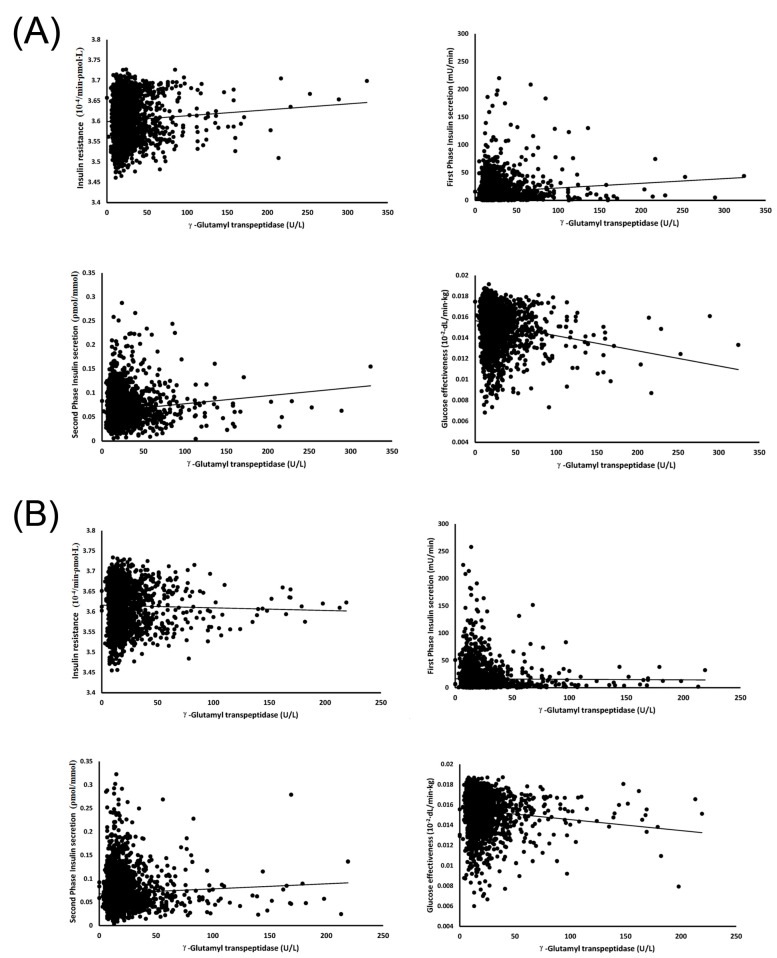
The scatter plots depict the relationships between γ−glutamyl transpeptidase and four diabetes factors (insulin resistance, first phase insulin secretion, second phase insulin secretion, and glucose effectiveness) for both men (**A**) and women (**B**), with accompanying regression lines. γ−glutamyl transpeptidase shows a significantly positive correlation with three of the diabetes factors (insulin resistance, first phase insulin secretion, second phase insulin secretion) but is negatively correlated with glucose effectiveness. However, in women, there is no significant relationship between γ−glutamyl transpeptidase and insulin resistance or first phase insulin secretion. The corresponding r and *p* values for each of these relationships are provided in Table 2.

**Figure 3 jcm-12-05606-f003:**
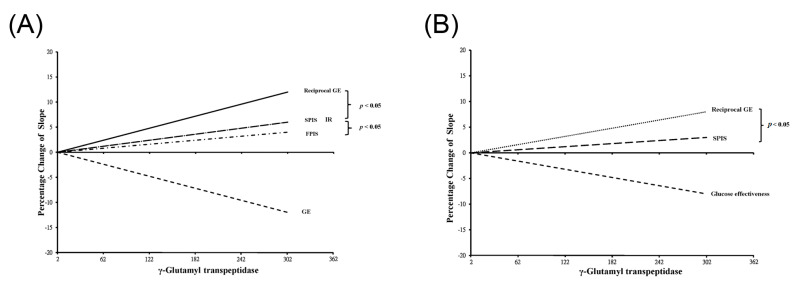
The comparison of the four regression lines of γ−glutamyl transpeptidase and diabetes factors in men (**A**) and women (**B**). These lines are presented as changes of the percentage of four slopes lines. Chris’s calculator was applied for the comparison of each correlation line. A steeper slope indicates a stronger relationship. The *p* values are shown at the right of the lines if there is significance between the two lines. Notably, in men, the slope lines for second phase insulin secretion (SPIS) and insulin resistance (IR) overlap. The slope for reciprocal glucose effectiveness (GE) is significantly steeper than that for SPIS and IR. However, the slope of the first phase insulin secretion (FPIS) is smoother than the other three diabetes factors in men. In panel B, the regression lines between γ−glutamyl transpeptidase, FPIS, and IR in women are not shown due to a lack of significant relationships. Nonetheless, the slope of reciprocal GE is significantly steeper than that of SPIS in women.

**Table 1 jcm-12-05606-t001:** The baseline demographic data and clinical characteristics of study participants.

	Men	Women
Number	2644	2598
Age (year) ***	65 ± 5.1	63 ± 4.0
Body mass index (kg/m^2^) *	23.0 ± 2.58	23.2 ± 2.83
Systolic blood pressure (mmHg) **	125.7 ± 17.54	126.9 ± 17.40
Diastolic blood pressure (mmHg) ***	74.6 ± 10.5	72.4 ± 9.98
Waist circumference (cm) ***	99.8 ± 14.7	97.8 ± 13.8
Glutamate oxaloacetic transaminase (µkat/L)	0.43 ± 0.20	0.42 ± 0.27
Glutamate pyruvate transaminase (µkat/L) ***	0.43 ± 0.30	0.39 ± 0.35
Creatinine (mg/dL) ***	1.1 ± 0.17	0.8 ± 0.1
Uric acid (mg/dL) ***	6.6 ± 1.4	5.4 ± 1.2
Triglycerides (mmol/L)	1.19 ± 0.55	1.20 ± 0.52
High density lipoprotein cholesterol (mmol/L) ***	1.36 ± 0.33	1.62 ± 0.37
Low density lipoprotein cholesterol (mmol/L) ***	3.29 ± 0.81	3.37 ± 0.87
γ-Glutamyl transpeptidase (U/L) ***	24.3 ± 21.8	19.1 ± 17.9

Data are presented as mean ± SD. * *p* < 0.05, ** *p* < 0.01, *** *p* < 0.001.

**Table 2 jcm-12-05606-t002:** Diabetes factors at the end of four-year follow up.

	Men (*n* = 2644)	Women (*n* = 2598)
First phase insulin secretion (μU/min)	16.0 ± 19.3	16.1 ± 21.4
Second phase insulin secretion (ρmol/mmol) ***	0.064 ± 0.032	0.070 ± 0.039
Insulin resistance (10^−4^/min∙ρmol∙L) ***	3.602 ± 0.052	3.615 ± 0.052
Glucose effectiveness (10^−2^∙dL/min∙kg) *	0.0153 ± 0.0018	0.0154 ± 0.0017

Data are presented as mean ± SD. * *p* < 0.05, *** *p* < 0.001.

**Table 3 jcm-12-05606-t003:** The results of Pearson’s correlation between baseline gamma-glutamyl transpeptidase and diabetes factors after four-year follow up.

	*r*	*p*
Men (*n* = 2644)		
First phase insulin secretion	0.103	<0.001
Second phase insulin secretion	0.119	<0.001
Insulin resistance	0.068	<0.001
Glucose effectiveness	−0.177	<0.001
Women (*n* = 2598)		
First phase insulin secretion	−0.008	0.696
Second phase insulin secretion	0.050	0.011
Insulin resistance	−0.021	0.276
Glucose effectiveness	−0.109	<0.001

## Data Availability

Data is available on request due to privacy/ethical restrictions.

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
