# Peer review of "Gamma-Glutamyltransferase Is a Predictor for Future Changes of Diabetogenic Factors in Aged Chinese—A Four-Year Follow-Up Study"

_jcm, 2023, doi:10.3390/jcm12175606_

Round 1
Reviewer 1 Report
1. The authors cite formula 1 and corresponds to model 1 of reference number 10. However, it does not match the formula of model 5 that obtained better correlation results in that reference. Could you justify this, please. Additionally, it is important to indicate how they transform (mathematical deduction) the NGT formula to insulin resistance.
2. For formula 3, the authors mention that this is the second phase of insulin secretion (SPIS), but in reference 13, this equation describes the deteriorated insulin secretion (ISEC). Please explain why it is considered the same.
3. The formulas are based on some parameters considered as pathologies included in metabolic syndrome, not only as risk factors for diabetes. It would be worth mentioning this limitation of prediction, because some diabetic patients have normal blood pressure and body mass index.
4. Why use the GE reciprocal value for the graphs in Figure 2. What is the mathematical justification and biochemical interpretation of this?
5. Triglycerides include quite a few molecules, for this reason, I suggest indicating this word in the plural form throughout the document (for example, in Table 1).
6. The discussion mentions: “Furthermore, we ranked the strength of these associations in descending order as GE, SPIS, FPIS, and IR. Despite this, the relationships between γGT and both FPIS and IR in women lacked statistical significance.” It would be important to discuss what the reasons for this are.
7. Perhaps they have yet to conclude on the effectiveness of formulas for predicting diabetes risks.
8. Perhaps it would be necessary to conclude the functionality/predictability of these formulas in the target population, highlighting the importance of this.
9. Please adhere to the authors' guide regarding references.
10. There are some drafting or presentation details marked in the attached document. For example:
a. Perhaps the activity of enzymes Glutamate oxaloacetic y Glutamate pyruvate transaminases should be expressed in international units.
b. Some symbols in the units of the graphs in Figure 1 may need to be corrected.

Author Response
We would like to express our gratitude to the reviewer for taking the time to review our article and providing valuable feedback. We have carefully considered the reviewer's recommendations and opinions. We appreciate the reviewer's efforts to help us improve the quality of our article. We would like to take the reviewer's questions and comments into account and have made the necessary corrections and revisions to the article. We have answered the reviewer's questions point to point and made sure that our article is now more accurate and informative.
Reviewer 1
- The authors cite formula 1 and corresponds to model 1 of reference number 10. However, it does not match the formula of model 5 that obtained better correlation results in that reference. Could you justify this, please. Additionally, it is important to indicate how they transform (mathematical deduction) the NGT formula to insulin resistance.
ANS: Thank you for the recommendation in the review. In our previous study, Model 5 showed the best correlation with IR. However, it requires parameters such as sex, age, and plasma glucose and insulin concentrations at 0, 30, 60, 90, 120, and 180 minutes during the Oral Glucose Tolerance Test (OGTT). Unfortunately, the majority of our subjects in the health examination did not undergo the OGTT. As a result, we lack all the necessary parameters for the current study. We only have data on fasting glucose and metabolic syndrome components. Consequently, although Model 5 is the best for assessing IR, it is unavailable for use in the study.
- For formula 3, the authors mention that this is the second phase of insulin secretion (SPIS), but in reference 13, this equation describes the deteriorated insulin secretion (ISEC). Please explain why it is considered the same.
ANS: We apologize for any confusion caused. In reference 12 (not 13) of our previous work, we published an original article where we estimated second phase insulin secretion, using the abbreviation "2nd ISEC" to denote this concept. However, in the current study, we have opted to use a different abbreviation, "SPIS," which represents the same concept of second phase insulin secretion. In summary, both "2nd ISEC" and "SPIS" refer to the same term.
- The formulas are based on some parameters considered as pathologies included in metabolic syndrome, not only as risk factors for diabetes. It would be worth mentioning this limitation of prediction, because some diabetic patients have normal blood pressure and body mass index.
ANS: We concur with the reviewer's suggestions. Several factors, aside from components of metabolic syndrome, contribute to the development of T2D, including family history and physical activity levels. It is important to note that some patients with T2D exhibit normal blood pressure and BMI. Furthermore, we have admitted the limitations of predicting T2D in the Discussion section.
- Why use the GE reciprocal value for the graphs in Figure 2. What is the mathematical justification and biochemical interpretation of this?
ANS: In the present study, we try to explore and rank the relationships among γGT and four DFs, ranging from strong to weak. We utilized Chris’s calculator for the comparison of each correlation lines. A steeper slope indicated a stronger relationship. Since GE is negatively correlated with γGT, we drew a reciprocal line of GE and placed the four DFs in the same quadrant for a comparative analysis of their slope steepness. With the change, readers are able to understand the relationship clearly.
- Triglycerides include quite a few molecules, for this reason, I suggest indicating this word in the plural form throughout the document (for example, in Table 1).
ANS: Thank you for the reviewer's suggestion. We have corrected all instances of "triglyceride" to the plural form, "triglycerides."
- The discussion mentions: “Furthermore, we ranked the strength of these associations in descending order as GE, SPIS, FPIS, and IR. Despite this, the relationships between γGT and both FPIS and IR in women lacked statistical significance.” It would be important to discuss what the reasons for this are.
ANS: We have attempted to elucidate the potential reasons for the observed differences in relationships between men and women. We have highlighted the roles of sex hormones and the differential distribution of fat between the sexes. The relevant discussion has been included in the Discussion Section.
- Perhaps they have yet to conclude on the effectiveness of formulas for predicting diabetes risks. Perhaps it would be necessary to conclude the functionality/predictability of these formulas in the target population, highlighting the importance of this.
ANS: Thanks reviewer’s suggestion. We added the importance in the Conclusion section.
- Please adhere to the authors' guide regarding references.
ANS: We have reviewed and corrected all references in accordance with the authors' guidelines.
- There are some drafting or presentation details marked in the attached document. For example:
- Perhaps the activity of enzymes Glutamate oxaloacetic y Glutamate pyruvate transaminases should be expressed in international units.
- Some symbols in the units of the graphs in Figure 1 may need to be corrected.
ANS: Thank you for the reviewer's suggestions. We have made all the corrections as recommended by the reviewer.
Once again, we would like to thank the reviewer for their time and effort in reviewing our article. Their comments and suggestions have been immensely helpful in improving the quality of our work. We hope that our revised article meets the high standards of your esteemed journal.
Sincerely,
Chung-Ze Wu M.D., PhD
Division of Endocrinology and Metabolism, Department of Internal Medicine, School of Medicine, College of Medicine, Taipei Medical University; No. 250, Wuxing St., Xinyi Dist., Taipei City 110, Taiwan
Tel: +886-2-22490088
E-mail: chungze@yahoo.com.tw
Reviewer 2 Report
The authors mentioned the interest of Gamma-glutamyltransferase as a predictor for future changes of
diabetogenic factors in aged Chinese.
The article is original in its content and the length of its development.
Some changes are worth mentioning:
- in the methodology, some explanations are important to mention to explain the use of these equations.
- how the population was sampled.
- how patients with hepatic impairment were eliminated: Ultrasound? Fibroscan ... knowing that these patients remain at risk of NASH because they are overweight or obese ...
- by explaining the mechanisms, γGT has been linked to fatty liver, which is
also associated with inflammation and IR, suggesting that fatty liver may serve as a bridge between γGT and IR. But in this case, the mechanism is actually NASH and hepatic steatosis, both of which were identified before the study began. In this case, it would be interesting to repeat the liver ultrasound explorations during follow-up?
- It's true that oxidative stress at the end explains everything, but in this case we're talking about allostatic load.
Allostatic load is the model that best explains the senescence of the organism. Allostatic load integrates several hormonal and metabolic models... For example, chronic cortisol elevation can alter liver function and increase GGT. This may explain the increased incidence of diabetes.
The authors can add this hypothesis to their discussion:
https://doi.org/10.1016/j.jcjd.2019.05.011
doi: 10.1016/j.psyneuen.2022.105841.
-That said, we don't see the implications of this study for clinical practice. If the allostatic load system explains biological wear and tear through its calculation, how can GGT help prevent or manage Diabetes early on?
Good
Author Response
We would like to express our gratitude to the reviewer for taking the time to review our article and providing valuable feedback. We have carefully considered the reviewer's recommendations and opinions. We appreciate the reviewer's efforts to help us improve the quality of our article. We would like to take the reviewer's questions and comments into account and have made the necessary corrections and revisions to the article. We have answered the reviewer's questions point to point and made sure that our article is now more accurate and informative.
Reviewer 2
- The authors mentioned the interest of Gamma-glutamyltransferase as a predictor for future changes of diabetogenic factors in aged Chinese. The article is original in its content and the length of its development. Some changes are worth mentioning:
In the methodology, some explanations are important to mention to explain the use of these equations. how the population was sampled.
ANS: Thanks for the reviewer’s suggestion. We have provided the population details for each formula in the Methods section.
- How patients with hepatic impairment were eliminated: Ultrasound? Fibroscan ... knowing that these patients remain at risk of NASH because they are overweight or obese
ANS: Thank you for the recommendation. We excluded subjects diagnosed with cirrhosis of the liver, chronic hepatitis B, or alcoholic hepatitis because these conditions can elevate γGT, potentially interfering with our results. However, we did not exclude those with fatty liver unless their AST/ALT levels were more than three times the upper limit. Consequently, we add the related descriptions in the Method Section on Page 2.
- By explaining the mechanisms, γGT has been linked to fatty liver, which is also associated with inflammation and IR, suggesting that fatty liver may serve as a bridge between γGT and IR. But in this case, the mechanism is actually NASH and hepatic steatosis, both of which were identified before the study began. In this case, it would be interesting to repeat the liver ultrasound explorations during follow-up?
ANS: We are grateful for the insights and suggestions provided by the reviewer. We wholeheartedly agree with the established relationship between hepatic steatosis and insulin resistance. Based on the prior research, it's evident that non-alcoholic fatty liver disease often presents with elevated levels of γGT. This correlation suggests that γGT could be more than just a casual marker; it may indeed serve as a predictive biomarker, indicating the early stages of fatty liver development and subsequent insulin resistance. It's important to note, however, that our study has a limitation: we did not have access to or incorporate data from abdominal ultrasounds. Recognizing the importance of this, we have taken care to highlight and discuss these limitations in the Discussion Section, in line with the suggestions you provided.
- It's true that oxidative stress at the end explains everything, but in this case we're talking about allostatic load. Allostatic load is the model that best explains the senescence of the organism. Allostatic load integrates several hormonal and metabolic models... For example, chronic cortisol elevation can alter liver function and increase GGT. This may explain the increased incidence of diabetes. The authors can add this hypothesis to their discussion: https://doi.org/10.1016/j.jcjd.2019.05.011; doi: 10.1016/j.psyneuen.2022.105841.
ANS: We sincerely appreciate the insights and recommendations offered by the reviewer. We concur with the proposed mechanism linking γGT to the four DFs through the allostatic load. Allostatic load represents the physiological strain from chronic stress. This stress elevates cortisol levels, disrupting metabolic balance and leading to IR. This resistance paves the way for type 2 diabetes. Stress-induced inflammation contributes to this metabolic dysregulation. We referenced the two articles and added the related descriptions in Discussion Section
- That said, we don't see the implications of this study for clinical practice. If the allostatic load system explains biological wear and tear through its calculation, how can GGT help prevent or manage Diabetes early on?
ANS: Thanks reviewer’s suggestion. We thoroughly discussed our results and explored possible explanations, thereby enhancing the clinical significance of our findings.
Once again, we would like to thank the reviewer for their time and effort in reviewing our article. Their comments and suggestions have been immensely helpful in improving the quality of our work. We hope that our revised article meets the high standards of your esteemed journal.
Sincerely,
Chung-Ze Wu M.D., PhD
Division of Endocrinology and Metabolism, Department of Internal Medicine, School of Medicine, College of Medicine, Taipei Medical University; No. 250, Wuxing St., Xinyi Dist., Taipei City 110, Taiwan
Tel: +886-2-22490088
E-mail: chungze@yahoo.com.tw
Reviewer 3 Report
The study of Wong and collegues aimed to investigate the relationship between γGT and four DFs in older Chinese individuals. The study was well designed and innovative, however I have some suggestion for the authors:
- there are some typos in the introduction. Please revise all the chapter.
- the authors should explain more in details why they used t-test and not the ANOVA for example for the statistical analysis
- please enlarge the text in all the figures because it is hardly readable
- please explain in the discussion clearly why and how these findings could be helpful for patients
Minor editing of English language is required. The reviewer detected some typos in the text, especially in the introduction chapter.
Author Response
We would like to express our gratitude to the reviewer for taking the time to review our article and providing valuable feedback. We have carefully considered the reviewer's recommendations and opinions. We appreciate the reviewer's efforts to help us improve the quality of our article. We would like to take the reviewer's questions and comments into account and have made the necessary corrections and revisions to the article. We have answered the reviewer's questions point to point and made sure that our article is now more accurate and informative.
Reviewer 3
- The study of Wong and collegues aimed to investigate the relationship between γGT and four DFs in older Chinese individuals. The study was well designed and innovative, however I have some suggestion for the authors:
There are some typos in the introduction. Please revise all the chapter.
ANS: Thank the reviewer's recommendations. We have corrected and removed our errors. We appreciate the time the reviewer took to assess our article, which has improved its readability.
- The authors should explain more in details why they used t-test and not the ANOVA for example for the statistical analysis
ANS: The t-test and one-way ANOVA are statistical methods for testing differences in means among groups. The t-test is used for comparing two groups, while one-way ANOVA is used for three or more groups. Both assume normal distribution and equal variances. In our study, since there were only two groups (men vs. women), we utilized the t-test for statistical analysis. Nonetheless, a steeper slope suggests a stronger relationship between γGT and the four DFs. In this scenario, neither the t-test nor one-way ANOVA is appropriate for statistical analysis. We used Chris's calculator to compare each correlation line.
- Please enlarge the text in all the figures because it is hardly readable
ANS: Thanks reviewer’s recommendation. We have provided detailed explanations for the figure legends, making it easier for readers to understand.
- Please explain in the discussion clearly why and how these findings could be helpful for patients
ANS: Thanks reviewer’s suggestion. We thoroughly discussed our results and explored possible explanations, thereby enhancing the clinical significance of our findings.
Once again, we would like to thank the reviewer for their time and effort in reviewing our article. Their comments and suggestions have been immensely helpful in improving the quality of our work. We hope that our revised article meets the high standards of your esteemed journal.
Sincerely,
Chung-Ze Wu M.D., PhD
Division of Endocrinology and Metabolism, Department of Internal Medicine, School of Medicine, College of Medicine, Taipei Medical University; No. 250, Wuxing St., Xinyi Dist., Taipei City 110, Taiwan
Tel: +886-2-22490088
E-mail: chungze@yahoo.com.tw
Round 2
Reviewer 2 Report
The article in its current form is acceptable for publication.
The authors made a great work in editing it.